# Fresh Pecorino Cheese Produced by Ewes Fed Silage with Prickly Pear By-Products: VOC, Chemical, and Sensory Characteristics Detected with a Neuro-Sensory Approach Combining EEG and TDS [note 1]

**DOI:** 10.3390/foods14193334

**Published:** 2025-09-25

**Authors:** Riccardo Gannuscio, Giuseppina Gifuni, Giuseppe Maniaci, David Bongiorno, Serena Indelicato, Claudia Lino, Marco Bilucaglia, Alessandro Fici, Margherita Zito, Vincenzo Russo, Massimo Todaro, Giuseppe Avellone

**Affiliations:** 1Department of Agricultural, Food and Forestry Science (SAAF), Università di Palermo, Viale delle Scienze 13, 90128 Palermo, Italy; riccardo.gannuscio@unipa.it (R.G.); giuseppe.maniaci@unipa.it (G.M.); 2Department of Business, Law, Economics and Consumer Behaviour “Carlo A. Ricciardi”, Università IULM, 20143 Milan, Italy; giuseppina.gifuni@studenti.iulm.it (G.G.); marco.bilucaglia@iulm.it (M.B.); alessandro.fici@iulm.it (A.F.); margherita.zito@iulm.it (M.Z.); vincenzo.russo@iulm.it (V.R.); 3Department of Biological, Chemical and Pharmaceutical Sciences and Technologies (STEBICEF), Università di Palermo, 90128 Palermo, Italy; david.bongiorno@unipa.it (D.B.); serena.indelicato@unipa.it (S.I.); claudia.lino@unipa.it (C.L.); beppe.avellone@unipa.it (G.A.); 4Behavior and Brain Lab IULM—Neuromarketing Research Center, Università IULM, 20143 Milan, Italy

**Keywords:** prickly pear by-products, silages, fresh pecorino cheese, VOC, neuromarketing, EEG, TDS

## Abstract

The reuse of by-products from plant processing as feed for animals aligns with the principles of a circular economy. Feeding dairy ruminants agro-industrial by-products often alters the chemical composition and sensory characteristics of dairy items. A dual approach—classic with neuro-sensory techniques—was utilized to evaluate the effect of prickly pear by-products on the diets of dairy ewes. Fresh Pecorino cheeses made from the milk of two groups of sheep fed with and without prickly pear by-product silage were analyzed for chemical composition and volatile organic compounds (VOCs). Furthermore, a neurosensory approach with consumers was used, combining electroencephalography (EEG) and temporal dominance of sensations techniques (TDS). Prickly pear silage in sheep diets did not alter the chemical composition of fresh cheese, but it did modify its fatty acids, with a significant increase in SFA (+2.60%) and PUFA (+0.33%), with a better n-6/n-3 ratio (−0.35%) due to higher omega-3 fatty acid content (+0.23%). The identification of VOCs revealed an increase in caproic acid (+27.27%) and n-caprylic acid (+6.47%) and a greater presence of sweet notes in the prickly pear-based cheeses, which exhibited a different aromatic complexity compared with the control cheeses. Even with a neuromarketing approach, sweetness remained the predominant sensation.

## 1. Introduction

In a world marked by continuous population growth and increasing environmental challenges, there is a strong focus on environmental protection, sustainable development, and animal welfare. Within this framework, agriculture and livestock farming are crucial, often cited as major contributors to environmental pollution [1]. Livestock farming is a major source of greenhouse gases, contributing approximately 14.5% of all human-induced emissions; is the main driver of freshwater use and pollution; and consumes 60% of the global biomass harvested annually to support human nutrition [1].

By 2050, it is projected that the demand for milk and meat will require approximately double the current production levels [2]. This highlights the critical responsibilities of agricultural and livestock operators. To reduce the ecological footprint of this sector and food–feed competition, it is essential to research new low-cost feeds that do not compete with humans [3]. A central strategy involves encouraging the reuse of plant processing by-products as livestock feed, in line with the principles of a low-impact circular economy. This approach emphasizes reducing reliance on external inputs, creating a closed-loop nutrient cycle, and minimizing the environmental burden of waste. The ultimate goal is to extract value from raw materials that would otherwise generate disposal costs [4].

The widespread availability of agro-industrial by-products (AIBPs) provides a strong incentive to explore innovative applications that move beyond their conventional functions. One particularly promising avenue lies in their incorporation into livestock diets, a strategy that not only promotes sustainability but also improves the overall efficiency of resource utilization while alleviating the competition for edible crops between human consumption and animal nutrition [5]. Many by-products are already included in ruminant diets, and interest is growing in identifying new options that offer potential nutraceutical benefits [6].

Therefore, particular attention is being given to identifying alternative solutions to this challenge. For instance, certain geographical areas are strongly associated with the cultivation of specific plants, such as the prickly pear (*Opuntia ficus-indica* L. Mill.) in Sicily (Italy) [7]. In these regions, initiatives are underway to make use of by-products generated from agro-industrial processing. The expansion of prickly pear cultivation in Sicily has, in fact, fueled a growing demand for fresh edible fruits and juice intended for human consumption. In 2024, 156,641 tons of fresh fruit were produced in Sicily, equal to 97.72% of national production, an increase compared with previous years [8]. Consequently, a substantial amount of prickly pear by-products (PPBs) comprising peel, pulp, and seeds following juice extraction is produced as waste [9]. To produce juice, the processing industry often grinds the entire fruit, including the skin, resulting in a residue known as “pastazzo”, which consists of skins, seeds, and residual pulp. To improve juice quality, automatic peeling machines are increasingly being used, enabling the efficient separation of peels, seeds (which are processed to obtain valuable oil), and juice/pulp [7]. This manufacturing process produces a considerable number of PPBs, which, if properly stabilized with ensiling, can constitute a valuable nutritional resource for small ruminants [10,11,12], particularly when fresh forage is unavailable. PPB silages represent a viable alternative feed for lactating ewes, as they provide a substantial source of digestible fiber and water. In addition, their inclusion in the diet helps lower feeding costs, enhances the sustainability of animal production, and positively influences milk composition by modulating rumen fermentation processes [13].

Moreover, these PPBs constitute a valuable source of bioactive molecules, including polyphenols and tannins [5]. When incorporated into sheep diets, such compounds can exert multiple positive effects, ranging from the optimization of rumen fermentation processes to the enhancement of animal health status, ultimately contributing to the production of dairy products with superior nutritional and qualitative attributes [14]. It is well known that the feed given to milk-producing ruminants often alters the volatile compounds in dairy animals [15,16]. Therefore, the ingredients in the feed, or their by-products, might be taken up by the animal during digestion, move to the mammary gland, and then end up in its milk [16]. When making cheese, some of these substances might impact the biochemical processes. This could happen either by interacting directly with the enzymes involved or by influencing the way bacteria’s genes are expressed [17]. Thus, studying the volatile organic compound (VOC) profile of a product can be very useful for establishing its sensory status; thus, this technique is widely used by researchers for fresh [18,19] and ripened cheeses [16,20].

In this context, understanding the market acceptance and consumer appreciation of these innovative products becomes equally crucial. According to market studies, each year, thousands of new food products are introduced to the market, yet a significant proportion fail shortly thereafter. Frequently, this market failure can be attributed to insufficient attention to consumers’ perceptions and hedonic responses toward these products. Traditionally, consumer acceptance studies have relied heavily on methods such as surveys, focus groups, and structured interviews. However, these approaches are often limited in accurately capturing true consumer preferences [21]. Contemporary neuroscientific insights have highlighted that humans are fundamentally emotional beings, influenced substantially by unconscious processes [21,22]. In this context, neuromarketing, an interdisciplinary domain integrating neuroscience, psychology, and marketing, offers a robust methodological approach to addressing these processes. In recent years, applied neuroscience techniques such as electroencephalography (EEG), galvanic skin response (GSR), and heart rate variability (HRV) have emerged as essential tools within food consumer science [21,23,24,25,26]. These techniques enable researchers to measure unconscious neurophysiological responses to food stimuli in real time, providing deeper insights into genuine consumer reactions. Additionally, sensory evaluation methods such as temporal dominance of sensations (TDS) offer a dynamic understanding of sensory perceptions by tracking the evolving dominance of specific sensory attributes throughout consumption. Unlike traditional sensory profiling, TDS captures the temporal evolution of taste and texture perceptions, making it particularly suitable for complex food matrices such as cheese, where multiple sensory attributes interact simultaneously [27,28]. A previous study effectively applied TDS to contexts involving multiple tastings, highlighting, for instance, how accompanying products like wine can modify sensory perceptions of cheese over consecutive bites [29]. Such methodologies are especially valuable for evaluating consumer reactions. Furthermore, understanding how product communication influences consumer acceptance is critical, a phenomenon well captured by the framing effect theorized by Kahneman [30]. Subtle priming, such as health-related cues, has been demonstrated to significantly alter food choices even without explicit consumer awareness, emphasizing the utility of neurophysiological measures in food decision-making research [25]. Moreover, prior research suggests emotional framing more effectively impacts sensory appreciation, whereas cognitive framing predominantly influences behavioral intentions [31]. Consequently, integrating TDS with neuromarketing techniques can provide comprehensive insights into both the affective and cognitive dimensions of consumer responses, informing more effective product communication strategies. This study proposes two important analysis steps on a specific consumer product, a special type of pecorino cheese, first analyzing the characteristics of the product itself and then the consumer’s perception through neuromarketing tools. More specifically, it aims to address two primary objectives through two specific steps:STEP 1: Evaluate the impact of prickly pear by-products as feed on the chemical characteristics and VOCs of fresh pecorino cheese through an experimental chemical analysis.STEP 2: Determine whether innovative cheeses elicit emotional and cognitive responses comparable to traditional cheeses through a neuro-sensory evaluation integrating EEG and TDS methodologies.

## 2. Materials and Methods

The experimental trial was conducted on a commercial farm in Menfi, located in Sicily’s Agrigento Province, Italy. The farm housed 500 Valle del Belice sheep bred using a semi-extensive system that involves grazing and supplementary feeding in a sheepfold. The trial followed the ethical principles of animal experimentation adopted by the Animal Welfare Commission of the University of Palermo (protocol number: UNPA-CLE 201954—12 December 2023).

### 2.1. Raw Material Collection and Ensiling

The PPS was supplied by a produce juice extraction company (Agres s.r.l., Carini, PA, Italy). The fruit was pressed whole in a press, the juice was separated, and the remaining “pastazzo” (pulp, peels, and seeds) was loaded onto trucks. After 24 h, it was sent to a livestock farm, where it was immediately ensiled by adding 12% wheat bran (relative to its unprocessed weight) for 50 days in airtight plastic containers with a degassing valve.

### 2.2. Animals, Experimental Design, and Ewe Feeding

A total of 250 lactating Valle del Belice ewes were divided into two experimental groups of 125, homogeneous in terms of parity and daily milk production. The ewe groups underwent a 2-week adaptation period to their new housing conditions and diet. At this time, both groups of ewes were pastured together. After the afternoon milking, they were separated. The experimental group received a daily supplement of prickly pear silage (PPS) (approximately 500 g/head) and as much sulla hay as they wanted; conversely, the control group was fed only sulla hay ad libitum. The gross composition of the PPS was as follows: dry matter (DM), 413.7 g/kg; crude protein, 95.5 g/kg; ether extract, 37.8 g/kg; neutral detergent fiber, 666.6 g/kg; non-fiber carbohydrate, 119.2 g/kg; ash, 80.8 g/kg; pH 4.04. That of the sulla hay was as follows: DM, 900.9 g/kg; crude protein, 71.5 g/kg; ether extract, 14.4 g/kg; neutral detergent fiber, 796.8 g/kg; non-fiber carbohydrate, 49.2 g/kg; ash, 68.2 g/kg.

Following the two-week adjustment phase, the mixed milk (from the evening and morning milkings) of each group was transformed, applying traditional “PDO Pecorino Siciliano” cheese technology [32]; two cheese-making processes were carried out weekly. The experimental plan included two different trials: CTR, control production that used the milk of ewes fed without PPS; EXP, experimental production with the milk of ewes fed with PPS. Briefly, bulk milk (200 L) heated at 38 °C was transferred into a 12-year-old Douglas wooden vat and maintained under gentle manual agitation for 5 min before the addition of lamb rennet paste (36 g; Rennet Regional Consortium, Poggioreale, Italy). Curdling occurred in approximately 40–50 min. The unbroken coagulum was mixed with hot water (20 L) at 74 °C to facilitate syneresis and was cut with a wooden paddle, called a “rotula”, until rice-sized grains (3–7 mm diameter) appeared. Once the whey was removed, the curd was manually pressed into 7 kg rattan baskets. This resulted in two cheeses: one for the EXP trial and one for the CTR trial. After the ricotta cheese was produced, the deproteinized whey (scotta whey) was used to cook the cheeses “under scotta”, that is, immersed in scotta whey at 72 °C for 3 h. After cooking, the cheeses were kept at room temperature in molds, and after 24 h of drying, they were salted in saturated brine for another 24 h and ripened for 7 days in a storage chamber at 12 °C and 85% of relative humidity (RH). After this, the cheeses were packaged under vacuum and stored at +5 °C until the analysis. The cheese production was carried out in duplicate after 1 week.

### 2.3. Milk and Cheese Analyses

Bulk milk samples were analyzed for lactose, fat, urea, and somatic cell count (SCC) using the infrared method (Combi-foss 6000, Foss Electric, Hillerød, Denmark). Total bacterial count (TBC) was determined with the BactoScan instrument (Foss Electric), pH was determined with an HI 9025 pH meter (Hanna Instruments, Ann Arbor, MI, USA), and titratable acidity was determined with the Soxhlet–Henkel method (SH/50 mL). Total nitrogen (TN), non-casein nitrogen (NCN), and non-protein nitrogen (NPN) were determined using standard FIL-IDF procedures [33,34] to calculate total protein (TN × 6.38) and casein [TN-(NCN × 0.994) × 6.38].

The 7-day cheese samples were lyophilized for successive analyses; DM, fat, protein (N × 6.38), and ash content were determined according to FIL-IDF procedures 4A [35], 5B [36], 25 [37], and 27 [38], respectively. Cheese samples were assessed and color measured in duplicate using a Minolta Chroma Meter CR300 (Minolta, Osaka, Japan) with illuminant C; the results are expressed as lightness (L*, from 0 = black to 100 = white), redness (a*, from red = +a to green = −a), and yellowness (b*, from yellow = +b to blue = −b), according to the CIE L* a* b* system [39]. Cheese hardness was evaluated with an Instron 5564 tester (Instron, Trezzano sul Naviglio, Milan, Italy), measuring the maximum resistance to compression (compressive stress, N/mm^2^) of samples (2 cm × 2 cm × 2 cm) kept at room temperature (22 °C).

Fatty acids (FAs) in freeze-dried cheese samples (100 mg) were directly methylated in 1 mL hexane with 2 mL of 0.5 M NaOCH_3_ at 50 °C for 15 min, followed by 1 mL of 5% HCl in methanol at 50 °C for 15 min, based on the bimethylation procedure described by Lee and Tweed [40]. Fatty acid methyl esters (FAMEs) were recovered in 1.5 mL of hexane. Using an autosampler, 1 μL of each sample was injected into an HP 6890 gas chromatography system equipped with a flame-ionization detector (Agilent Technologies, Santa Clara, CA, USA). FAMEs from each sample were separated using a CP-Sil 88 capillary column (100 m long, 0.25 mm internal diameter, 0.25 µm film thickness) (Chrompack, Middelburg, The Netherlands). The injector temperature was held at 255 °C, and the detector temperature was held at 250 °C, with a hydrogen flow of 40 mL/min, an air flow of 400 mL/min, and a constant helium flow of 45 mL/min. The initial oven temperature was held at 70 °C for 1 min, increased by 5 °C/min to 100 °C, held for 2 min, increased by 10 °C/min to 175 °C, held for 40 min, then finally increased by 5 °C/min to a final temperature of 225 °C, held for 45 min. Helium was used as carrier gas, with 158.6 kPa of pressure and a flow rate of 0.7 mL/min (linear velocity 14 cm/s). A FAME hexane mix solution (Nu-Check-Prep, Elysian, MN, USA) was used to identify each FA.

### 2.4. Solid-Phase Microextraction of Cheese Volatile Components

Six grams of the sample were placed in a 40 mL glass vial with a silicon septum. The prepared sample was placed in a thermostatic water bath and equilibrated for 30 min at 60 °C. Volatile compounds were extracted by exposing a 50/30 μm divinylbenzene–carboxen–polydimethylsiloxane (DVB/CAR/PDMS) fiber (Supelco, Bellefonte, PA, USA) to the sample headspace for 30 min. Following extraction, the solid-phase microextraction (SPME) fiber was withdrawn, inserted into the injector port, and thermally desorbed at 230 °C for 5 min. Before use, the fiber was conditioned in a GC injector at 270 °C for 1 h.

The SPME-GC/MS experiments were carried out by performing GC/MS analysis on the volatile compounds released in the headspace of a closed vial upon SPME using a thin polymer coating fixed to the solid surface of a DVB/CAR/PDMS fiber. A Thermo Fisher Scientific (Waltham, MA, USA) TSQ 800 triple quadrupole mass spectrometer and a Thermo Fisher Scientific Trace 1310 gas chromatminograph analyzer (Waltham, MA, USA) were used for the GC/MS analyses. The analytical conditions used appear in Table 1.

Samples were analyzed using a TG XLBMS column (20 m × 0.18 mm i.d. × 0.18 μm, Thermo Scientific GC Column), and the injector port was held at 230 °C. The column carrier gas was helium (99.999%) at a constant flow rate of 1.2 mL/min. The oven temperature was held at 35 °C for 5 min, increased to 100 °C at a rate of 5 °C/min, held for 2 min, and then increased to 180 °C at a rate of 6 °C/min and maintained for 2 min. Finally, it was increased to 230 °C at the rate of 8 °C/min and held for 2 min. The mass analysis was carried out under electronic ionization (EI) conditions with an ionization energy of 70 eV, and the source temperature was set at 300 °C. Full-scan acquisition was used in the 30 to 350 m/z range. Each component was identified by comparison with the mass spectra of the NIST library. Each determination was repeated three times.

### 2.5. Statistical Analysis

Data on milk and cheese composition were analyzed using a one-way analysis of variance (ANOVA) model [41]. The following statistical model was applied:(1)Y_ik_ = μ + D_i_ + ε_ik_ where Y_ik_ is the dependent variable, µ is the general average, D_i_ denotes the fixed effect of the i Diet (i = CTR and EXP), and ɛ_ik_ is the residual error. When a statistically significant effect (*p*-value < 0.05) was detected, the means were compared using Student’s *t*-test at 0.01 and 0.05 *p*-levels.

### 2.6. Neuro-Sensory Approach

This study employed a within-subject design integrating neurophysiological, sensory, and self-report measures to investigate the cognitive–emotional and sensory responses elicited by an innovative upcycled cheese (EXP cheese) compared with a traditional counterpart (CTR cheese).

#### 2.6.1. Sample

The participants were recruited via a specialized external agency, which ensured an initial selection based on predefined inclusion criteria. In total, 27 subjects (10 males, 17 females) between 20 and 59 years old (mean: 38, standard deviation: 38 ± 13.47) participated in the experiment; however, the final sample was reduced to 23 (9 males, 14 females) after excluding incomplete or poor-quality EEG data. Inclusion criteria included the absence of dairy allergies and minimal familiarity with cheese consumption, which were assessed using a preliminary questionnaire. Mean age was balanced across genders (mean: 40, standard deviation: 38 ± 13.25; t(21) = 0.69, *p* = 0.49), while the proportion was not (χ^2^(1) = 1.08, *p* = 0.29). Furthermore, the analysis of consumption habits showed that 81% of the sample eat cheese regularly, claiming to consume it two to three times a week; moreover, 15% of the sample eat cheese once a week, while 4% claim to consume it daily.

All participants provided written informed consent before taking part in this study. The consent forms were signed by each participant and a research team member and securely stored in accordance with local Data Protection Regulations.

#### 2.6.2. Instruments

EEG data were recorded using an NVX-36 device (Medical Computer Systems Ltd., Moscow, Russia) with 22 Ag/AgCl electrodes evenly placed at standard 10-10 scalp locations (Fp1, Fp2, F7, F3, Fz, F4, F8, T3, C3, Cz, C4, T4, T5, P3, Pz, P4, T6, Po3, Po4, O1, Oz, and O2) plus 2 Ag/AgCl earclips attached to the left (A1) and right (A2) earlobes. The montage was monopolar, internally grounded, and referenced to one adhesive Ag/AgCl patch placed on the right mastoid (M2). Recordings were performed at 2000 Hz/24 bits and controlled by the NeoRec software (Medical Computer Systems Ltd.).

Prior to electrode placement, the skin was prepared using NuPrep scrubbing gel and Neurgel conductive paste (Spes Medica, S.r.l., Battipaglia, Italy), ensuring impedance levels below 10 kΩ, in line with standard guidelines [42].

With regard to the sensory and taste component, the temporal dominance of sensations (TDS) method was implemented, which consists of asking participants to indicate, in real time via a special keyboard, the dominant taste sensation that emerges during a predetermined 30 s time frame. A special TDS keyboard was developed for this study with four attributes (sweet, sour, bitter, savory), presented in a fixed order. The sensations investigated were defined based on the fresh Sicilian Pecorino cheese data sheet drawn up by the ONAF organization. The analysis of the frequency–dominance curves obtained from the TDS made it possible to represent the temporal evolution of the sensations perceived, thus providing a more detailed picture of the sensory dynamics than can be obtained using traditional static methods.

Finally, a set of self-report questionnaires was administered to collect both subjective and socio-demographic information, as well as data on consumption habits (such as frequency of purchase and average expenditure) and purchasing propensity. After each tasting, perceptions were collected through a battery of standardized questionnaires measuring perceived quality (6-point Likert scale) and hedonic liking (10-point scale). The combination of these measurements, integrated and compared with neurophysiological data and the results of the TDS analysis, made it possible to comprehensively and multidimensionally examine consumers’ reactions to the products tested.

#### 2.6.3. Data Processing

EEG data preprocessing was conducted using MATLAB Version 9.10 (The MathWorks, Inc., Natick, MA, USA) and the EEGLAB toolbox. The raw signal was re-referenced to the linked earlobes, A1–A2, and resampled to 512 Hz. A band-pass filter (0.1–40 Hz zero-phased IV order Butterworth filter, Charlottesville, VA, USA) and the CleanLine [43] regression-based multi-taper method (at both 50 and 100 Hz) were applied first. Then, non-stationary artifacts (e.g., movements, and cable swinging) were corrected using the Artifact Subspace Reconstruction method [44]. The Independent Component Analysis (ICA), based on the FastICA algorithm [45], was applied, and stereotypical artifacts (e.g., muscle noise and ocular movements) were automatically removed using ICLabel [46]. To speed up the ICA convergence, the weight matrix was computed based on a downsampled (128 Hz) and heavily filtered (1–30 Hz IV order zero-phased Butterworth filter) copy of the data [47]. To enhance the spatial resolution at the sensor level, the signal was finally re-referenced to the Current Source Density [48].

From the eye-closed resting state epoch, the power spectral densities (PSDs) of the occipital channels were estimated according to the Welch’s method (1 s long hamming window with 50% overlapping) and averaged. Then, the Individual Alpha Frequency was estimated as the center of gravity of the average PSD within the 7.5–12 Hz band.

The Approach–Withdrawal Index (AWI) was computed as the difference in alpha power (IAF ± 2 Hz) between the right (F4, F8) and left (F3, F7) prefrontal channels. The Individual Alpha Frequency (IAF) was determined for each participant using Welch’s method, based on the power spectral density of occipital channels during a resting state with eyes closed. Derived from prefrontal alpha activity, the Approach–Withdrawal Index (AWI) is a neurophysiological indicator of affective predisposition toward a stimulus: higher values indicate a positive emotional orientation and greater implicit attractiveness. In the context of the present study, the increase in AWI observed during the tasting of the innovative cheese—particularly under the post-priming condition—can be attributed to its specific taste characteristics, such as the greater perception of sweetness and lower saltiness, which were also highlighted by the TDS analysis. These sensory attributes are associated with more rewarding taste experiences and may activate neural patterns [49,50] compatible with a positive affective response. This suggests that changes in the volatile and lipid composition of cheese, derived from a sheep’s diet, can modulate not only conscious perception but also the automatic emotional processing of the product. Statistical analyses were conducted based on EEG and TDS data using repeated-measures ANOVA, paired-samples t-tests, and post hoc comparisons.

#### 2.6.4. Experimental Design

The experiment took place at the Behavior and Brain LAB of IULM University in Milan.

The experimental design was based on two main conditions of tasting the same product, organized with the objective of assessing in the absence of information and then in the presence of information, priming the sensory and emotional–cognitive perception of the participants. Before acquiring the experimental data, all participants underwent a brief TDS training session to allow the subjects to familiarize themselves with the interface and the concept of “dominant sensation” before the four official tasting phases began [28].

The first condition involved a “blind” tasting, in which subjects tasted the two sheep’s milk cheeses in analyses without receiving any prior indication as to their nature or production methodology.

The second condition included the priming component, i.e., communication aimed at enhancing the sustainability or territoriality aspects of the cheeses, in accordance with information manipulation procedures previously experimented with by Jaeger and MacFie [51]. In this phase, participants were provided with information on the innovation and “upcycling” of the product for the EXP cheese and the territorial origin and classic method used for the CTR cheese. Under both conditions, EEG data were collected in conjunction with the temporal dominance of sensations exercise to simultaneously examine the dynamics of sensations and neurophysiological responses.

The cheeses were presented in rectangular molds weighing 5 g each [52,53]. To reduce any habit or taste persistence effects, the order of presentation of the two kinds of cheese was randomized, and between each tasting, participants took a sip of water to neutralize the palate [53]. At the end of each tasting, they were asked to fill in questionnaires on enjoyment and perceived quality to collect subjective judgments to be compared with the results obtained from instrumental measurements. Overall, each experimental session lasted forty minutes per person. This study was conducted in accordance with the Declaration of Helsinki and was approved by the Ethics Committee of IULM University (approval n. 0067812).

## 3. Results and Discussion

### 3.1. Chemical Composition

The composition of the milk from the sheep in the two feeding groups is shown in Table 2. Significant differences were found in the percentages of fat (*p* < 0.01), casein (*p* < 0.05), and urea content (*p* < 0.01), which was higher in the milk from the control group.

Altogether, the milk parameters align with the average values of the Valle del Belice sheep, though urea content was higher [54]. These lower fat and casein percentages in the EXP milk could be due to the higher daily milk yield of EXP ewes, as a result of the supplementation of prickly pear silage. This hypothesis is supported by the higher lactose content (*p* < 0.10) found in the milk of the EXP group. In fact, there is a known positive correlation between the percentage of lactose in milk and daily milk production [55]. The high urea content found in both milk groups could be due to the period of investigation, January and February, which is characterized by pastures with an abundance of young, green grass that increases dietary protein intake [56]. The significantly lower urea content of the EXP milk is probably due to the effect of diet supplementation with prickly pear silage, which is richer in non-structural carbohydrates [12] that balance the protein–carbohydrate ratio.

The parameters shown in Table 3 align with the average chemical composition reported for Pecorino Siciliano PDO cheese [32], but no significant differences were found between groups. The hardness values are typical of fresh cheese, with lower results than those reported for ripened Pecorino Siciliano cheeses [32]. Regarding color indices, no significant differences were found between the cheeses of the two groups, which showed the typical parameters of fresh Sicilian Pecorino cheese [32,57].

Cheese FA composition is reported in Table 4. It is widely recognized that diet plays a major role in modulating the FA composition of sheep milk [58] and, consequently, cheeses. The majority of FAs belonged to the saturated FA class (on average, 66.20% and 68.80% in the CTR and EXP groups, respectively).

These data are consistent with those reported for Pecorino Siciliano PDO cheese [59]. In particular, palmitic and myristic acids among the saturated acids and oleic acid among the unsaturated acids were the most abundant fatty acids, aligning with the literature [59,60]. Supplementing the sheep diet with prickly pear silage partially modified the fatty acid profile of the cheese. The saturated FAs of the cheeses produced with milk from the EXP group were significantly higher than those from the CTR group (*p* < 0.01), mainly because of the increase in the two main fatty acids of the cheese, myristic acid and palmitic acid. The increase in palmitic acid in the EXP cheese is probably due to the prickly pear supplementation; indeed, this acid is rather abundant in prickly pear pulp and peels [61,62]. Palmitic acid may not have been modified in the rumen environment but rather has been partially lost in the rumen. It could not completely reach the small intestine, where it is absorbed and transferred to the blood and, consequently, milk [63].

Prickly pear by-products also contain high levels of unsaturated fatty acids, especially oleic and linoleic acids [61,62]. However, grazing plays a key role in improving the health benefits of milk and dairy products from sheep [64] and cows [65] for humans, as it increases monounsaturated and polyunsaturated fatty acids, which are more abundant in fresh forages. However, the fatty acid composition of milk greatly depends on the ruminal biohydrogenation of polyunsaturated FA and the enzymatic activity of Δ9-desaturase in the mammary gland [[66],]. Khiaosa-Ard et al. [67] found that dietary phenols could inhibit the last step of ruminal biohydrogenation of linoleic and linolenic fatty acids, leading to their increase, in agreement with Cifuni et al. (2023) [68] but in contrast to Capucci et al. [69], who found that supplementing dairy ewe diets with different doses of phenolic concentrate did not affect CLA or vaccenic acid concentrations. Therefore, the inconsistency of the results might be related to the number of phenols fed and the ratio of individual compounds present in the extract. The negative or beneficial effects of phenols on the biohydrogenation of dietary PUFA may vary depending on the composition of the basal diet, the source of phenols, and the amount included in the diet [14].

The cheeses obtained from the milk of the experimental group did not show a high percentage of oleic or linoleic acid, probably because of the high intake of pasture grass, which, being prevalent over prickly pear silage, likely masked the expected effects. Conversely, alpha-linolenic acid (ALA) was significantly higher in the cheeses of the experimental group, resulting in more omega-3 fatty acids with a better n-6/n-3 ratio.

More than 30 total volatile compounds were identified in the cheese samples, representing the aromatic profiles of samples analyzed using HS-SPME-GC/MS. Compounds were classified into chemical groups for each sample: alcohols, carbonylic compounds, esters, fatty acids, terpenes, and hydrocarbons (Table 5). For each compound, the relative abundance was evaluated, though this quantitative approach has limitations, as the areas of each peak do not reflect the real quantities of the different compounds. However, these percentages are very useful as a comparison tool and provide useful indications when evaluating the contribution of prickly pear by-products to the flavor profiles of cheeses.

In the CTR cheese sample, 23 compounds from various classes were identified, including 2 alcohols, 4 carbonyl compounds, 8 esters, 2 fatty acids, 1 terpene, and 6 hydrocarbons (Table 6). The most relevant alcohol identified was ethanol, a by-product of microbial fermentation, which imparts sweet notes to cheese [70]. Among carbonyl compounds, nonanal and dodecanal were detected in lower concentrations as products of lipid oxidation, contributing floral, citrus-like, and fatty notes that enhance the complexity of the flavor profile [70]. Additionally, 2-heptanone was also identified; together with δ-octalactone, it contributes sweet, fruity, and peach-like notes, arising from the hydrolysis and subsequent lactonization of hydroxylated fatty acid triglycerides during the ripening process [71].

Among the most abundant compounds detected were esters, which are associated with floral and fruity notes [70]. These compounds likely contribute to the cheese’s aroma by balancing the acidity of fatty acids and reducing the bitterness of amines [72]. Esters are primarily formed through enzymatic or chemical reactions between fatty acids and primary alcohols [73]. They can also result from the transesterification of partial glycerides [74]. Capric acid was identified as the most abundant compound, accounting for 32.11% of the relative area.

The presence of citronellol in cheese samples is likely due to the animal feeding on fresh forage and native pasture plants, many of which are naturally rich in terpenoid compounds such as citronellol and its precursors. These compounds can be transferred from plants to milk and persist or undergo transformation during cheese ripening [75]. This finding underscores the significant influence of animal diet—particularly the intake of fresh, aromatic forage—on the development of flavor profiles in cheese [75].

In the experimental cheese produced with the milk of ewes supplemented with prickly pear silage, 28 compounds from various classes were identified, including 1 alcohol, 5 carbonyl compounds, 8 esters, 4 fatty acids, 1 terpen, and 9 hydrocarbons. It is interesting to note that EXP cheese contained several compounds that were absent in the CTR cheese, suggesting that they originate from the prickly pear by-products. The key compounds included 2-undecanone, nonanoic acid, and several hydrocarbons. Furthermore, the relative percentages of some compounds increased from the CTR to EXP cheeses, indicating that supplementing the ewe diets with prickly pear by-products significantly affected the volatile flavor profile of the cheese. Specifically, caproic acid increased from 32.11% to 59.38%, and n-caprylic acid increased from 4.48% to 10.95%. These fatty acids are known to contribute fruity, acidic, and creamy characteristics to the cheese’s aroma and flavor [76]. This increase suggests that adding prickly pear by-products to the diets of lactating ewes likely influenced the microbial fermentation process or the biochemical reactions involved in lipolysis, thus improving the flavor complexity of the cheese [77] without the appearance of off-flavors. These results suggest that prickly pear by-product silage can be partially incorporated into dairy ewe diets to reduce feeding costs and feed–food competition [13].

**Table 6 foods-14-03334-t006:** Volatile aromatic compounds expressed as a percentage of three replicates expressed as (peak area of each compound/total area of significant peaks) × 100, and odor descriptors identified in samples CTR and EXP cheeses.

n.	CAS	Chemical Compounds	Cheese (Mean ± s.e.)		Aroma	Odor Threshold μg/kg in Water	Reference
			CTR	EXP			
		**Alcohols**					
1	64-17-5	Ethanol	14.72 ± 0.90 ^a^	7.89 ± 1.23 ^b^	sweet	100,000	[70,71]
2	543-49-7	2-Heptanol	0.04 ± 0.01	n.d.	Citrus, Earth, Fried, Mushroom, Oil	70	[78,79]
		**Carbonylic Compounds**					
3	124-19-6	Nonanal	0.08 ± 0.03	0.07 ± 0.03	Fat, Floral, Green, Lemon	1	[70,71]
4	112-54-9	Dodecanal	0.06 ± 0.03	0.05 ± 0.04	Citrus, Fat, Lily	-	[70,71]
5	110-43-0	2-Heptanone	1.10 ± 0.05	0.98 ± 0.33	Penetrating fruity odor	140–3000	[70,71]
6	112-12-9	2-Undecanone	n.d.	0.11 ± 0.02	Fresh, Green, Orange, Rose	6.2	[78,79]
7	698-76-0	δ-octalactone	0.07 ± 0.02	0.07 ± 0.01	sweet, fruity, and peach-like	400	[79,80]
		**Esters**					
8	105-54-4	Ethyl butyrate	4.31 ± 0.47 ^a^	1.77 ± 0.25 ^b^	Sweet, Fruit, Cognac	-	[70,71]
9	106-70-7	Methyl caproate	1.14 ± 0.09 ^a^	0.64 ± 0.12 ^b^	Ester, Fresh, Fruit, Pineapple	70–84	[70,71]
10	123-66-0	Ethyl caproate	7.69 ± 0.55	3.41 ± 0.21	Fruity	1	[70,71]
11	111-11-5	Methyl caprylate	0.18 ± 0.05 ^a^	0.13 ± 0.01 ^b^	Fruit, Orange, Wax, Wine	-	
12	106-32-1	Ethyl caprylate	2.60 ± 0.13	1.30 ± 0.01	Fruit, fat	-	[70,71]
13	110-42-9	Methyl caprate	0.08 ± 0.01	0.09 ± 0.01	Fruit, fat	-	
14	110-38-3	Ethyl caprate	0.98 ± 0.09	0.45 ± 0.18	Brandy, Grape, Pear	-	[70,71]
15	106-33-2	Ethyl laurate	0.02 ± 0.01	0.01 ± 0.01	Floral, Fruit, Leaf	400	[78,79]
		**Fatty acids**					
16	142-62-1	Caproic acid	32.11 ± 3.43 ^b^	59.38 ± 0.62 ^a^	Cheese, Oil, Pungent, Sour	0.0006	http://www.leffingwell.com/odorthre accessed on 15 June 2025
17	124-07-2	n-caprylic acid	4.48 ± 0.41 ^b^	10.95 ± 0.45 ^a^	Cheese, Fat, Grass, Oil		
18	112-05-0	Nonanoic acid	n.d.	0.09 ± 0.09	Fat, Green, Sour		
19	334-48-5	Capric acid	n.d.	0.51 ± 0.14	Rancid		
		**Terpenes**					
20	1117-61-9	Citronellol	0.33 ± 0.02	0.34 ± 0.01	Citrus, Green, Rose		[75,76]
		**Hydrocarbons**					
21	1632-16-2	Heptane, 3 methylene	0.11 ± 0.01	n.d.			
22	111-84-2	Nonane	0.07 ± 0.01	n.d.	Gasoline-like odor		
23	17615-91-7	Undecane 5,6-dimethyl	n.d.	1.88 ± 0.06	Gasoline-like odor		
24	17,301-94-9	Nonane 4-methyl	n.d.	0.43 ± 0.06	Gasoline-like odor		
25	5911 4 6	Nonane 3-methyl	n.d.	2.39 ± 0.18	Gasoline-like odor		
26	13,475-82-6	Heptane, 2,2,4,6,6 pentamethyl	25.82 ± 2.35	4.36 ± 0.05	Gasoline-like odor		
27	124-18-5	n-Decane	0.23 ± 0.07	0.68 ± 0.02	Gasoline-like odor		
28	62,183-79-3	2,2,4,4 Tetramethyloctane	2.28 ± 0.05	0.27 ± 0.12	Gasoline-like odor		
29	17,301-23-4	Undecane 2,6 methyl	n.d.	0.28 ± 0.09	Gasoline-like odor		
30	629-50-5	Tridecane	n.d.	0.17 ± 0.01	Gasoline-like to odorless		
31	629-62-9	Pentadecane	0.06 ± 0.01	0.09 ± 0.03	Oil of *D. guineense* fruit		
32		Unknow	1.42 ± 0.01	1.21 ± 0.01	-		

CTR: control diet (hay and pasture); EXP: experimental diet (prickly pear silage, hay and pasture). On the row: different letters are significant for *p* < 0.05. n.d. = not detected

### 3.2. Neuro-Sensory Approach

To test whether innovative and traditional cheeses were perceived similarly from an emotional–cognitive point of view, the AWI was examined in the absence of any product information (blind condition). The analysis (repeated-measures ANOVA) showed that the mean AWI values did not differ significantly between the two kinds of cheese (F (1,22) = 1.42, *p* = 0.247, η^2^p = 0.060), confirming H1: under information-neutral conditions, the upcycled product does not arouse different emotional reactions from the traditional one. This result aligns with previous studies [79], which state that if the basic sensory characteristics are superimposable, the absence of information reduces the potential bias related to novelty and sustainability.

The increase in specific volatile compounds such as capric and decenoic acid, detected in cheese produced from sheep fed with prickly pear silage, is known to impart pungent, creamy, and slightly acidic aromatic notes [71]; these characteristics are reflected in the TDS curves, where the sensation of sweetness appears dominant and persistent, while the saltiness is attenuated compared with traditional cheese. This change in the sensory profile, which is more balanced, smoother, and less salty, is consistent with the EEG data, which show an increase in the AWI after informative priming, suggesting an implicitly more positive emotional response, probably favored by the greater pleasantness associated with sweetness and the sustainable narrative of the product. Although the increase in short-chain fatty acids in EXP cheeses—known for imparting pungent and creamy notes [75]—is reflected by the greater intensity of sweetness and lower saltiness detected in the TDS, suggesting a direct biochemical–perceptual translation, the analysis revealed differences between the “blind” and “post-priming” phases.

The comparison between the “blind” and “post-priming” conditions showed a significant increase in AWI (*p* < 0.05) for both cheeses, although it was more pronounced in the CTR cheese. The main effect of the condition (blind vs. post-priming) suggests that, in general, there is a cognitive difference between the “blind” and “post-priming” conditions: irrespective of the type of cheese, semantic priming significantly affects cognitive perception.

This evidence corroborates H2, namely, that by highlighting aspects of sustainability and “upcycling”, semantic priming amplifies the tendency toward an emotional–cognitive approach to perception [23,80]. As a result, when consumers are informed about the environmental benefits or the peculiar production method, their implicit attitudes toward the product tend to improve.

However, the non-significance of the interaction between the condition and type of cheese indicates that the condition effect (blind vs. post-priming) does not vary significantly depending on the specific cheese. In other words, while the general condition (blind vs. post-priming) impacts perception, this impact does not depend on the type of cheese. Although the cheese–condition interaction was not significant in the ANOVA, the post hoc and t-test results suggest that there is a positive trend toward a different and potentially more positive perception after priming.

The first hypothesis postulated that, in the absence of information, the perception curves obtained from the TDS would not show significant divergences between the cheeses if they were perceived as equally hedonic. The results, however, partially disprove H1: the analysis (paired samples) reveals substantial differences in some key sensations. In particular, in the CTR cheese, the sensations “bitter” and “savory” reach temporally different and more pronounced dominance values than in the EXP cheese, with *p* < 0.01 at different observation times. These findings suggest that the participants, despite having no information, picked up intrinsic sensory differences, likely attributable to specific production characteristics or raw materials [24].

The cheeses are perceived significantly differently (F(1120) = 66.756, *p* < 0.001, η^2^p = 0.357).Each cheese induces a different sensation profile (F(3360) = 44.392, *p* < 0.001, η^2^p = 0.270).

Comparing the blind and post-priming conditions, significant differences were found in perception curves, also confirming the effect of semantic priming on dynamic sensations. The information provided altered the sequence and intensity of perceived sensations, highlighting how communication and information context can alter not only general appreciation but also the detailed perception of sensory features.

Cheese perception curves vary significantly depending on the experimental condition (F (1120) = 5.278, *p* = 0.023, η^2^p = 0.042).The way in which the sensations are perceived changes between the two conditions (F (3360) = 5.462, *p* = 0.001, η^2^p = 0.044).

Post hoc analyses confirmed the differences (at *p* < 0.001).

Priming significantly impacts the sensations of bitterness and flavor. In fact, after priming, the EXP cheese was perceived as more bitter and less savory. Sweetness remained the predominant sensation in both conditions.

With the introduction of information, the comparison between blind and post-priming conditions showed marked variations for some sensations: overall, EXP cheese appears more dynamic and subject to sensory reinterpretation when participants receive information on the production methodology. This confirms H2, which aligns with previous research [20] emphasizing how communication can influence the perceptual construction of flavors.

It follows that adequate communication about the benefits and peculiarities of a product can alter the temporal sequence in which taste sensations emerge, which is in line with the literature on the framing effect [81].

The sensations of “bitterness” and “sweetness” did not differ significantly in terms of perception between the two cheeses. Sweetness is again significantly more perceived for the innovative cheese. At the same time, the CTR cheese is significantly more savory.

The results demonstrate that using agro-industrial by-products from prickly pears as feed for sheep not only is a sustainable strategy for biomass reuse but also leads to measurable changes in the volatile properties and sensory profiles of cheese, without compromising consumer appreciation. Integrating chemical–sensory analysis and neurophysiological measurements has shown that the alteration of the aromatic profile—particularly related to the increase in short-chain fatty acids—corresponds to a greater dominance of sweetness and a more positive affective response (AWI), especially in the presence of sustainability-oriented communication messages.

The combined EEG + TDS approach, still little used in food research, has proved particularly effective in exploring the complexity of the consumer experience, combining temporal perception and implicit response. This integrated protocol represents a replicable model for the analysis of new circular or functional products, helping to overcome the psychological barriers associated with the adoption of more sustainable food solutions and promoting marketing strategies based on neuroscientific evidence.

Moreover, the use of priming can open an important discussion on product communication, with important consequences for producers, as our results show how this can influence sensorial perceptions and implicit attitudes toward products when consumers are informed about production methods and benefits to the environment. This is in line with studies suggesting that information about the characteristics of a product, its production method, its origin, and its territorial history can strongly impact the choice of product and the intention to buy it [82]. Furthermore, this information can have an important impact on product safety perceptions (from nutrients to healthy ingredients), influencing consumer choices [26].

## 4. Conclusions

Supplementing sheep diets with prickly pear silage modified some aspects of the chemical composition of milk and cheese. Among the macro-chemical constituents of milk, only the urea content decreased, while no differences were recorded for the other parameters. Consequently, the chemical composition of the cheeses was not modified by supplementing the ewe diets with prickly pear silage. The fatty acid composition of the experimental cheeses showed higher SFA values because of increased palmitic acid and PUFA, with a healthy increase in omega-3 fatty acids.

The identification of VOCs highlighted the greater presence of sweet notes in the experimental cheeses, which present a different aromatic complexity from the control cheeses. Even with the neuromarketing approach, sweetness remained the predominant sensation in both experimental conditions, evidently thanks to integrating the sheep diets with prickly pear silage.

Moreover, these results confirm that a cheese obtained from upcycling flows can compete—on an implicit and explicit hedonic level—with a traditional product, especially when any communication emphasizes the environmental benefits. This evidence creates dialogue with two strands in the literature: on the one hand, the effects of expectations in sensory perception, and on the other hand, interventions in the corporate food environment, showing how supply-side modifications and “choice architecture” promote healthier and more sustainable purchases.

Furthermore, perceived sustainability and message clarity appear to be key levers in steering consumers toward more sustainable options. Our work confirms that information framing (priming) enhances the emotional–cognitive approach measured at the EEG level and recalibrates the dynamic sensory profile.

This study has some limitations. First of all, the relatively small sample size adopted in the neuro-sensory approach reduces its statistical power and limits the generalizability of the results to the wider population. Replication with larger and more heterogeneous groups of consumers is needed to confirm the robustness of the findings. However, despite these limitations, our findings can help companies design campaigns (e.g., green labels, supply chain storytelling, and visual nudging) and training on environmental impacts, enhancing coherence between brand narratives and consumer behavior. The joint use of EEG and TDS, still uncommon in food science, has strong potential, as it enables parallel analysis of sensory dynamics and implicit neuro-emotional responses. This approach provides a replicable model for studying innovative or sustainable products, offering a deeper, multidimensional understanding of consumer experience.

## Figures and Tables

**Table 1 foods-14-03334-t001:** Conditions employed for the SPME-GC/MS analysis.

Sample/Matrix	Fresh Pecorino Cheese “Primo Sale”
SPME fiber	50/30 µm DVB/CAR/PDMS
Sample equilibration	30 min, into a thermostatic water bath at 60 °C
Extraction	30 min, into a thermostatic water bath at 60 °C
Column	TG XLBMS column, L = 20 m × I.D. = 0.18 mm × df = 0.18 μm (Thermo Scientific GC Column)
Injection T	230 °C
Detector	Triple quadruple
Scan range	Full scan, *m*/*z* 30–350
Carrier gas	He 99.9999%, 1.2 mL/min

**Table 2 foods-14-03334-t002:** Bulk milk composition.

Items	Diet		SEM	*p*-Value
	CTR	EXP		
Fat (%)	6.04	5.48	0.151	0.001
Protein (%)	5.83	5.79	0.102	0.106
Casein (%)	4.47	4.39	0.113	0.035
Urea (mg/dl)	46.71	44.31	0.135	0.001
Lactose (%)	4.25	4.28	0.022	0.098
pH	6.71	6.78	0.028	0.238
Titratable acidity (SH/50 mL)	5.01	4.81	0.150	0.005
Somatic Cell Count (Log10/mL)	6.14	6.27	0.118	0.012
Total Bacterial Count (Log10/mL)	5.62	5.44	0.025	0.039

CTR: control diet (hay and pasture); EXP: experimental diet (prickly pear silage, hay and pasture). SEM: Standard Error of Mean.

**Table 3 foods-14-03334-t003:** Cheese chemical composition.

Items	Diet		SEM	*p*-Value
	CTR	EXP		
Dry matter (DM, %)	61.4	63.1	0.43	0.109
Fat (% DM)	43.2	44.1	0.09	0.110
Protein (% DM)	45.3	45.6	0.38	0.676
Ash (% DM)	11.2	10.1	0.09	0.181
Hardness (N/mm^2^)	0.393	0.389	0.054	0.966
Lightness, L*	75.27	73.49	0.715	0.221
Redness, a*	−5.79	−5.91	0.456	0.875
Yellowness, b*	18.49	16.72	1.668	0.531

CTR: control diet (hay and pasture); EXP: experimental diet (prickly pear silage, hay and pasture). SEM: Standard Error of Mean.

**Table 4 foods-14-03334-t004:** The effect of diet on fatty acid (FA) composition (g/100 g FA) of cheese fat.

Fatty Acids (FA)	Diet		SEM	*p*-Value
	CTR	EXP		
Total FA, % DM	45.64	47.11	1.407	0.488
C4:0	2.95	2.93	0.222	0.963
C6:0	3.00	3.13	0.208	0.687
C8:0	2.89	3.07	0.163	0.465
C10:0	7.64 ^B^	8.51 ^A^	0.260	0.050
C10:1	0.37 ^B^	0.41 ^A^	0.011	0.050
C12:0	4.06 ^B^	4.51 ^A^	0.021	0.001
C12:1	0.14 ^B^	0.17 ^A^	0.001	0.001
C14:0	10.67 ^b^	11.39 ^a^	0.176	0.028
C14:0 iso	0.09 ^B^	0.10 ^A^	0.002	0.007
C14:1	0.75	0.78	0.014	0.154
C15:0	0.94 ^B^	1.04 ^A^	0.017	0.006
C15:0 iso	0.22 ^a^	0.21 ^b^	0.004	0.050
C15:1	0.19	0.21	0.003	0.089
C16:0	24.10 ^b^	25.41 ^a^	0.368	0.045
C16:1	1.97	1.61	0.286	0.407
C17:0	0.59	0.61	0.007	0.073
C17:1	0.15	0.16	0.003	0.457
C18:0	8.02 ^A^	6.89 ^B^	0.027	0.001
C18:1 t11, TVA	3.71	3.67	0.018	0.206
C18:1 c9	14.39 ^A^	11.90 ^B^	0.071	0.001
C18:1 others	4.49	4.37	0.051	0.156
C18:2 n 6, LA	1.65	1.64	0.016	0.611
C18:2 c9 t11 CLA	1.68 ^A^	1.59 ^B^	0.009	0.005
C18:3 n-6	0.06	0.06	0.001	0.330
C18:3 n-3, ALA	1.01 ^B^	1.26 ^A^	0.008	0.001
C19:0	0.46	0.49	0.019	0.351
C20:0	0.24 ^A^	0.18 ^B^	0.003	0.001
C20:4 n-6, AA	0.09	0.08	0.002	0.213
C20:5 n-3, EPA	0.09	0.09	0.001	0.105
C21:0	0.05 ^B^	0.06 ^A^	0.001	0.001
C22:0	0.11 ^A^	0.09 ^B^	0.004	0.010
Saturated FA	66.20 ^B^	68.80 ^A^	0.374	0.002
Monounsaturated FA	27.63 ^A^	24.69 ^B^	0.370	0.001
Polyunsaturated FA	6.18 ^B^	6.51 ^A^	0.051	0.003
n-6 PUFA	1.98 ^a^	1.93 ^b^	0.014	0.045
n-3 PUFA	1.11 ^A^	1.34 ^B^	0.008	0.001
n-6/n-3	1.79 ^A^	1.44 ^B^	0.010	0.001

CTR: control diet (hay and pasture); EXP: experimental diet (prickly pear silage, hay and pasture). SEM: Standard Error of Mean. On the row different superscript letters are significant: ^a, b^ = *p* < 0.05; ^A, B^ = *p* < 0.01. TVA = trans vaccenic acid. LA = linoleic acid. CLA = conjugated linoleic acid. ALA = α-linolenic acid. AA = arachidonic acid. EPA = eicosapentaenoic acid. PUFA = polyunsaturated fatty acids.

**Table 5 foods-14-03334-t005:** Volatile compounds identified in the cheese samples (Area %).

Chemical Compounds	DIET	
	CTR	EXP
Alcohols	14.76	7.89
Carbonylic Compounds (Ketones and Aldehydes)	1.30	1.29
Esters	17.01	7.80
Fatty acids	36.60	70.93
Hydrocarbons	28.57	10.54
Terpenes	0.33	0.34
Unknow	1.42	1.21

CTR: control diet (hay and pasture); EXP: experimental diet (prickly pear silage, hay and pasture).

## Data Availability

The original contributions presented in this study are included in the article. Further inquiries can be directed to the corresponding author.

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
