# Peer review of "Fresh Pecorino Cheese Produced by Ewes Fed Silage with Prickly Pear By-Products: VOC, Chemical, and Sensory Characteristics Detected with a Neuro-Sensory Approach Combining EEG and TDSâ€"

_foods, 2025, doi:10.3390/foods14193334_

Round 1
Reviewer 1 Report
Comments and Suggestions for Authors
Thanks for inviting me to review the manuscript titled “Fresh Pecorino Cheese produced by ewes fed with silage of prickly pear by-products: VOC, Chemical and Sensory Characteristics detected with a Neuro-Sensory Approach combining EEG and TDS”. The involvement of prickly pear seems interesting. I would asked authors to address questions as followed:
- The abstract should focus on the findings of the experiments in this study with critical stats and explanation. The second paragraph was more like a supplement to background which should be in “introduction” part.
- Line 81-82, it seems lack of connections between two paragraphs. The describe of prickly pear by-products was abrupted in the middle of somewhere. The “valuable nutritional resource” should be extended.
- Section 2.2, “About 250” ewes was recruited and divided into 2 groups, is there an exact number of how many ewes were involved rather than a blurry description?
- Section 3.2, the results mentioned that “the pleasantness of the EXP cheese was more pronounced” but did not provide statistical support for this claim in the blind condition.
- Ethical Approval: The IRB approval number is provided, but the name of the ethics committee is missing.
- The conclusion: the authors failed to discuss the limitation of the study and further studies.
Author Response
see you attached file

Reviewer 2 Report
Comments and Suggestions for Authors
The manuscript is well-written. Here are a few points to consider for revision.
Line 23-41: Some of the short names are not familiar to readers, so you may need to expand those short names.
Lines 177-179: It is a little confusing, after cooking for 3 hours, how the cheese was held before salting.
Line 364: Can you also report the chemical composition of the Feed? It would be helpful to understand the transition of feed to milk.
Line 397: Can you report PDO Pecorino Siciliano cheese data along with your taste data for comparison?
Line 500: What do you mean by rounder in sensory? Please explain.
Discussion: Could you please also discuss the cost difference between feed using PPS and conventional feed?
Conclusion: What are the future research scope and limitations of your study?
Author Response
see you attached file

Reviewer 3 Report
Comments and Suggestions for Authors
This review examines a dual approach, classic and with the neuro-sensory techniques, was utilized to evaluate the effect of prickly-pear by-products in diet of dairy ewes. It explores the effects of feed ingredients or their by-products on dairy products from ruminant dairy animals, with a focus on prickly pear silage. It analyzes fresh Pecorino cheeses made from the milk of sheep fed with and without prickly pear by-product silage for chemical composition and VOCs and uses a combination of EEG and TDS technology to evaluate the effect of sheep diet on cheese flavors detected by consumers. The findings highlight that the chemical composition of the cheeses was not modified by the supplementation of the ewe diet with prickly pear silage. The fatty acid composition of the experimental cheeses showed higher values of SFA due to the increase in palmitic acid and PUFA with a healthy increase in omega-3 fatty acids. The identification of VOCs highlight a greater presence of sweet notes in the experimental cheeses, which present a different aromatic complexity compared to the control cheeses. The experimental results also confirmed that the information framework (activation) enhanced the emotion-cognition approach measured at the EEG level and recalibrated the dynamic sensory profile. Companies can translate this evidence into targeted campaigns (green labels, supply chain storytelling, visual nudging) and training sessions explaining the environmental impact, thus increasing the coherence between brand narratives and consumers' everyday behavior. These results help confirm that combining EEG and TDS can be used to study the temporal dynamics of sensory perception and implicit neuro-emotional responses to be investigated in parallel. This combination offers a replicable model for the study of innovative or sustainable products, providing a deeper and more multidimensional understanding of the consumer experience.
However, I therefore have to point out some comments:
Line 45-46: The description " Within this framework, agriculture and livestock farming are crucial, often cited as major contributors to environmental pollution " is too vague. In order to illustrate the connection between agriculture and livestock farming and environmental pollution, please specify the main aspects or provide specific data showing that these activities cause pollution.
Line 50-52: The mention of " One promising strategy involves innovating in animal nutrition using new ingredients, additives, and feeding practices, moving beyond traditional methods " lacks of connection with the preceding text.
Line 57-59: The description " Without their incorporation into livestock feed, these by-products could significantly impact the environment due to their increasing volume, further exacerbated by rising population and consumption of processed foods" lacks of supporting literature.
Line 70-72: The claim " The cultivation of prickly pear in the Sicilian region has increased the demand for fresh edible fruits or juice for human consumption in recent years" lacks specific details. Please specify the changes in human demand and the source of this data to improve credibility.
Line 83-85: The description "Therefore, the ingredients in the feed, or their by-products, might be taken up by the animal during digestion, move to the mammary gland, and then end up in the milk" lacks of supporting literature. Please add relevant references to make the article more credible.
Line 85-86: The mention of "When making cheese, some of these substances might impact the biochemical processes" is too general. Please specify the specific categories of substances that affect biochemical processes and provide the necessary references.
Line 110-112: The claim "In recent years, applied neuroscience techniques such as electroencephalography (EEG), galvanic skin response (GSR), and heart rate variability (HRV) have emerged as essential tools within food consumer science" lacks of necessary references.
Line 147-151: The Materials and Methods section should include the specific location where fresh prickly pear by-products were harvested or the harvesting company.
Line 488-489: Simply stating “the mean AWI values did not differ significantly between the two kinds of cheese” is insufficient. Please provide statistical details, such as p-values or correlation coefficients, to support this conclusion.
Comments on the Quality of English Language
no
Author Response
see you attached file

Reviewer 4 Report
Comments and Suggestions for Authors
Dear editor
I have made a preliminary review of the manuscript. After checking the Ithenticate report, I found a serious and high level of plagiarism in the manuscript. For example, please check that most of introduction and material and methods were textually copied from other already published sources (see attached files). Under these circumstances, a manuscript should be rejected without review. I do not review a manuscript with high levels of plagiarism. Sorry.
Under these circumstances, I recommend the rejection of the manuscript without review to maintain the quality standards of the journal.

High plagiarism, I recommend the rejection.
Author Response
We regret the hasty conclusions of the Reviewer, but the analysis of plagiarism with software often does not take into account that the partially identical parts of the text are often methods of other papers of ours already published and do not concern either the results or the discussions of the paper. However, we have slightly modified the text of the methods.
Reviewer 5 Report
Comments and Suggestions for Authors
The authors have submitted an article that outlines an interesting explorative study on fresh Pecorino cheese produced by ewes fed with silage of prickly pear by-products. The manuscript is interesting and fits in the scope of the Foods.
The title and keywords accurately reflect the content of the manuscript. The abstract should be corrected as suggested in the submitted PDF file. The introduction section is too long and references are missing in many sentences. Research aim of the study is clearly defined. The materials and methods are described clearly with sufficient details of the performed measurements and the measurement techniques are appropriate to resolve the stated objectives of the study. However, there are some information that have to be added. Also, statistical approach should be clarified and better explained. The obtained results are well presented in an unbiased, detailed, clear and easily comparable manner where you can clearly draw the conclusion. There are no unnecessary data presented. From the discussion the conclusion can be easily extrapolated. Conclusion section is too long and represents just a summary of the obtained results, and, thus, should be corrected. Even though my mother language in not English, manuscript should be checked by native speaker, because there are some confusing sentences in the manuscript. References consist of appropriate and relevant papers.
I do believe this work is worthy of publication in Foods journal, but I would recommend a major changes before it is published. My comments and suggestions are outlined in the submitted PDF file.

Author Response
see you attached file

Round 2
Reviewer 2 Report
Comments and Suggestions for Authors
Author has responded to all the suggestions given to revise the manuscript. Revised manuscript looks good to me. I do not have any more comment for improvement.
Author Response
We thank the Reviewer for his support.
Reviewer 4 Report
Comments and Suggestions for Authors
Dear Editor,
Sorry, but as I said in my previous review, I found a high similarity of this manuscript with other sources. The authors stated that this high similarity is only with a thesis. However, methods from other papers were also textually included in this manuscript. Both practices are not tolerated in several well-recognized journals, and I agree with that point of view. Hence, I cannot recommend this manuscript for publication.
Considering that the editor decided to accept the revision of this manuscript, I would like to point out that all responsibility for future problems regarding the aforementioned manuscript relays in the Editorial Team of the journal.
Thank you for considering me to review this manuscript. However, please do not send me future submissions with these particular problems. In general, highly respected journals rejects/suggest corrections in manuscripts with this condition, before sending to review.
Author Response
Thank you very much for your valuable feedback and for taking the time to evaluate our manuscript. We acknowledge your concerns and, following the Editor’s guidance, we have revised the introduction section to address the requested modifications.
Reviewer 5 Report
Comments and Suggestions for Authors
Authors significantly improved the quality of the manuscript after incorporation of the reviewers suggestions. Therefore, it should be accepted for publication as it stands.
Author Response
We thank the Reviewer for his support.